# Graph Neural Networks and Arithmetic Circuits

**Timon Barlag**
Institute for Theoretical Computer Science
Leibniz University Hanover
Hanover, Germany
`barlag@thi.uni-hannover.de`

**Vivian Holzapfel**
Institute for Theoretical Computer Science
Leibniz University Hanover
Hanover, Germany
`holzapfel@thi.uni-hannover.de`

**Laura Strieker**
Institute for Theoretical Computer Science
Leibniz University Hanover
Hanover, Germany
`strieker@thi.uni-hannover.de`

**Jonni Virtema**
School of Computer Science
University of Sheffield
Sheffield, United Kingdom
`j.t.virtema@sheffield.ac.uk`

**Heribert Vollmer**
Institute for Theoretical Computer Science
Leibniz University Hanover
Hanover, Germany
`vollmer@thi.uni-hannover.de`

## Abstract

We characterize the computational power of neural networks that follow the graph neural network (GNN) architecture, not restricted to aggregate-combine GNNs or other particular types. We establish an exact correspondence between the expressivity of GNNs using diverse activation functions and arithmetic circuits over real numbers. In our results the activation function of the network becomes a gate type in the circuit. Our result holds for families of constant depth circuits and networks, both uniformly and non-uniformly, for all common activation functions.

## 1 Introduction

Neural networks have recently received growing attention from a theoretical point of view in a number of papers studying their computational properties. Relevant to this paper are examinations of the computational power of neural networks after training, i.e., the training process is not taken into account but instead the computational power of an optimally trained network is studied. Starting already in the nineties, the expressive power of feed-forward neural networks (FNNs) has been related to Boolean threshold circuits, see, e.g., [Maass et al., 1991, Siegelmann and Sontag, 1995, Maass, 1997, He and Papakonstantinou, 2022]. Most importantly, Maass [1997] showed that when restricting networks to Boolean inputs, a language can be decided by a family of (in a certain sense) "polynomial-size" FNNs if and only if it belongs to the class $TC^0$, i.e., the class of all languages decidable by families of Boolean circuits of constant depth and polynomial size using negation gates and unbounded fan-in AND, OR, and threshold gates.

In the last five years attention has shifted to the study of *graph neural networks (GNNs)* which is a model for machine learning tasks on graph-structured inputs. Their expressive power can be closely related to the Weisfeiler-Leman algorithm [Xu et al., 2019, Morris et al., 2019, 2021]. Originally developed to solve the graph isomorphism problem, the WL algorithm is connected to fragments of first-order logic; hence the logical expressiveness of GNNs was subsequently studied. Barceló et al.

38th Conference on Neural Information Processing Systems (NeurIPS 2024).

[2020] consider so-called logical classifiers—these are unary formulas of first-order (FO) predicate logic that classify a node in a given graph according to whether the formula holds for this node. Barceló et al. proved that all classifiers definable in a certain fragment GC (guarded first-order logic with counting quantifiers) are computable by GNNs. The converse direction is a bit problematic; it is only known to hold for unary queries definable in first-order logic. This result has been broadened recently by [Benedikt et al., 2024] using logics with Presburger quantifiers. Barceló et al. then consider an extension of GNNs by so-called global readout and show that these GNNs can compute all queries from the logic $FOC_2$ (first-order restricted to two variables, but extended with counting quantifiers, a superclass of GC), but the converse is open.

**Our motivation.** Neural networks in general and GNNs specifically have mostly been studied through the lens of Boolean functions. This does not necessarily reflect on the usage of machine learning models in real-world applications. We want to shift this focus to real-valued computations and hence, in this paper, we make real numbers first-class citizens. When networks are digitally simulated in practical applications, of course only rational numbers are used as inputs, and the real numbers that appear during their computation (e.g., via the sigmoid function) are approximated by rationals. However, we are interested in principal statements about the computational power of neural networks—hence we study networks as devices operating with real numbers. We do not solely consider networks that are restricted to Boolean inputs or Boolean outputs (Boolean queries, logical classifiers), as done in all the papers cited above. Instead, we consider GNNs as a model to compute functions from (vectors of) real numbers to (vectors of) real numbers, or from labeled graphs (i.e., undirected graphs whose nodes are annotated with vectors of real numbers) to labeled graphs.

Instead of turning to logics, we focus on a computation model that has been used to capture the expressivity of neural networks in the past: we use circuits, but since we turn away from the Boolean computation model, we use *arithmetic circuits* over the reals, that is, circuits that take real numbers as inputs and have nodes that compute real functions such as addition, multiplication, projection, or a constant function. Why do we do that? It will turn out that in this way we obtain a close correspondence between GNNs and arithmetic circuits. Arithmetic circuits can in a second step be simulated by a Boolean computation model, but this does not say anything about the computational power of the networks. Going directly from neural networks to Boolean circuits mixes up two different issues and obscures statements about the power of GNNs. By separating the two aspects, we do not only obtain an upper bound, or correspondence with respect to discrete classification tasks, but an equivalence between GNNs and constant-depth arithmetic circuits over the real functions they compute. In this way, we shed more light on the computational model behind GNNs. Our results show very explicitly what the computational abilities of GNNs are and what elementary operations they can perform.

We also want to make a general statement about scaling and complexity. A common narrative nowadays seems to be that by making neural networks larger and larger, they can solve more complex reasoning tasks. However, theoretical limitations as proven in this paper show that scaling is not all we need. Our results show that to improve the expressivity of GNNs, more computationally complex aggregation and combine functions are needed. More precisely, these functions need to come outside of the class $FAC^0_{\mathbb{R}}$, which contains functions that can be computed by arithmetic circuits that are bounded in size and depth, see Definition 2.11 for a precise definition. Another approach would be to change the fixed depth framework of GNNs to one that includes recursion, for example.

Following a broad line of research outlined by Merrill [2022], in this paper, we aim at a general statement about the computational power of neural networks following the GNN architecture, i.e., networks operating in layers where connections within layers and hence their communication capabilities correspond to the given input graph. However, contrary to most predecessor papers [Morris et al., 2019, Barceló et al., 2020], we do not only consider so-called AC-GNNs, i.e., GNNs where the computation in each node of a layer of the network consists first of an aggregation step (collecting information from the adjacent nodes of the graph—very often simply summation), followed by a combine step (combining the aggregated information with the current information of the node—mostly a linear function followed by a unary non-linear activation function such as sigmoid or ReLU). AC-GNNs like these are just one example of a GNN architecture. We take a more general approach. The GNN framework defines a continuous graph-based "message passing" [Gilmer et al., 2017, Morris et al., 2019]. In our networks we keep the layout and message passing mechanisms of GNNs but equip the nodes in each layer with constant-depth arithmetic circuits (taken from a previously fixed basis of circuits), not limited to the usual aggregate-combine (AC) operations. We call these networks

C-GNNs, circuit graph neural networks. AC-GNNs, whose aggregation function is computable by a constant-depth circuit, can be simulated by C-GNNs (using the same activation function). The computational upper bounds we obtain thus point out general computational limitations for networks following the GNN message passing framework.

**Main Results.** The main contribution of our paper is an exact correspondence between C-GNNs and arithmetic circuits. A function (from labeled graphs to labeled graphs) is computable by a C-GNN with a constant number of layers if and only if it is computable by a constant-depth arithmetic circuit over the real numbers. The activation function of the neural network will be a gate type in the arithmetic circuit. Thus, the (set of) actual activation function(s) becomes a parameter in a general statement equating the computational power of GNNs and arithmetic circuits.

A number of remarks are in order.

- Our result holds for all commonly used activation functions.
- Our result is uniform. If we start with a family of GNNs whose individual networks can be generated via some algorithm, the resulting family of arithmetic circuits will be uniform in the same way, and vice versa.
- Our result gives general limits for neural networks following the GNN message passing framework, including but not restricted to the currently widely used AC-GNNs. This means that in order to use a GNN to compute a function not computable by constant-depth arithmetic circuits, scaling GNNs up or adding computational power in the individual nodes will not help, but neural networks of a completely different architecture are needed.

This paper follows a modular approach. We stick to the GNN message passing framework, but allow computations of arbitrary power in the individual nodes of the graph. Our approach is adaptable and opens possibilities for many extensions; we point out a few in our conclusion section.

We would like to stress that theoretical insights like those presented in this paper have the potential to guide practical development. We prove principal limits of networks following the GNN framework, extending the current AC-GNNs. Our results enables the use of theoretical results regarding arithmetic circuit complexity for arguing about the complexity of GNNs. For example findings for a complexity hierarchy within the real valued $\text{FAC}_{\mathbb{R}}^0$ will lead to corresponding statements about the power of GNNs. They may also be used to design GNN architectures with provable expressivity bounds.

**Organization.** In the next section, we introduce the relevant notions about GNNs and arithmetic circuits. Section 3 contains our results. In Subsection 3.1, we introduce Circuit-GNNs as a generalization of AC-GNNs. In Subsections 3.2 and 3.3, we show how to simulate C-GNNs by arithmetic circuits, and vice versa. In Section 4, we point out a number of questions for further research. Due to lack of space, we only provide proof sketches for some of the proofs and refer to the arXiv version Barlag et al. [2024] for the full proofs.

**Related Work.** The first model of GNNs was introduced by Scarselli et al. [2009]. Since then, their expressive power has been studied in numerous works and from various perspectives as discussed above. As mentioned already, Barceló et al. [2020] established a close connection between classifiers definable in guarded $\text{FOC}_2$ and computable by AC-GNNs. It is worth to point out that a connection very close to the aforementioned one can be obtained by connecting the results of Sato et al. [2019], that relate graph neural networks with more general models of distributed computation, to the results of Hella et al. [2015], that logically characterize expressivity of those models of distributed computation with graded modal logics (known to be equivalent with guarded $\text{FOC}_2$). Grohe [2023] very recently extended these characterization results for GNNs, and furthermore connected GNNs to Boolean circuits. He considers an FO-fragment GFO+C (guarded fragment of FO plus a more general form of counting than used by Barceló et al.) and proves equivalence of this logic to GNNs for unary queries, that is for Boolean functions (functions with Boolean output). However, his overall result holds only in the non-uniform setting (that is, the sequence of GNNs for graphs of different sizes is non-uniform in the sense that there is not necessarily an algorithm that, given a graph size, computes that network from the sequence responsible for the given size), and the activation functions have to be "rpl approximable" (i.e., rational piecewise linear approximable, i. e., the real functions must in a certain way be a approximable by Boolean circuits or logic). This includes most of the commonly used activation functions. Since queries expressible in this logic are known to be $\text{TC}^0$ computable, he obtains a $\text{TC}^0$ upper bound for the computational power of GNNs. For the converse connection, an extension of GNNs is necessary, and it is shown that a unary query computable by a

TC$^0$ circuit is computable by a GNN with so-called random initialization. Another line of work that focuses on real-valued computations includes [Geerts et al., 2022] where the logic MPLang operating directly on real numbers was introduced to readily express GNN computations. The main goal of that paper was to reason about GNNs using a logic, extended by real-number computations. However, the converse connection, from MPLang to GNNs, is highly unclear. Other work addressing the precise characterizations of expressivity issues of GNNs outside of the restriction to the Boolean setting with GNNs using formal logic include [Cucala et al., 2023, Pfluger et al., 2024].To the best of our knowledge, there are no other works considering real-valued computations and circuits or arithmetic circuits. Another approach that leans more into comparing the expressivity when using different functions for aggregation in the GNN can be found in Rosenbluth et al. [2023], where they compared the expressivity of GNNs using sum as their aggregation with mean and max GNNs.

## 2 Preliminaries

Throughout this paper, the graphs we consider have an ordered set of vertices and are undirected unless otherwise specified. We use overlined letters to denote tuples, write $|\overline{x}|$ to denote the length of $\overline{x}$ (i.e., the number of elements of $\overline{x}$), and use $[n]$ to denote the first $n$ nonzero natural numbers $\{1, \ldots, n\}$. For a $k \in \mathbb{N}$ we denote all possible $n$-tuples for every $n$ of $\mathbb{R}^k$ with $\left(\mathbb{R}^k\right)^*$. The notation $\{\!\!\{\}\!\!\}$ is used for multisets. A restriction of a function $f$ to set $S$ is written as $f{\restriction}_S$.

**Definition 2.1.** Let $k \in \mathbb{N}$, and let $G = (V, E)$ be a graph with an ordered set of vertices $V$ and a set of undirected edges $E$ on $V$. Let $g_V : V \to \mathbb{R}^k$ be a function which labels the vertices with attribute vectors. We then call $\mathfrak{G} = (V, E, g_V)$ a *labeled graph of dimension $k$* and denote by $\mathfrak{Graph}$ the set of all labeled graphs and by $\mathfrak{Graph}_k$ the set of all labeled graphs of dimension $k$.

For a node $v \in V$, the *neighborhood of $v$* is defined as $N_{\mathfrak{G}}(v) \coloneqq \{w \in V \mid \{v, w\} \in E\}$.

In order to compare standard graph neural networks and the circuit graph neural networks which we will introduce later on, we fix the notion of a graph neural network first. Generally, GNNs can classify either individual nodes or whole graphs. For the purpose of this paper, we will only introduce node classification as graph classification works analogously. We focus on aggregate combine graph neural networks that aggregate the information of every neighbor of a node and then combine this information with the information of the node itself.

An *aggregation function* (of dimension $k$) is a permutation-invariant function $\text{AGG} : \mathbb{R}^k \times \cdots \times \mathbb{R}^k \to \mathbb{R}^k$, a *combine function* (of dimension $k$) is a function $\text{COM} : \mathbb{R}^k \times \mathbb{R}^k \to \mathbb{R}^k$ and a *classification function* (of dimension $k$) is a function $\text{CLS} : \mathbb{R}^k \to \{0, 1\}$ that classifies a real vector as either true or false. Finally, an *activation function* (of dimension $k$) is a componentwise function $\sigma : \mathbb{R}^k \to \mathbb{R}^k$.

**Definition 2.2** (AC-GNN, cf. [Barceló et al., 2020])**.** An $L$ layer *aggregate combine graph neural network* (AC-GNN) is a tuple $\mathcal{D} = (\{\text{AGG}^{(i)}\}_{i=1}^L, \{\text{COM}^{(i)}\}_{i=1}^L, \{\sigma^{(i)}\}_{i=1}^L, \text{CLS})$, where $\{\text{AGG}^{(i)}\}_{i=1}^L$ and $\{\text{COM}^{(i)}\}_{i=1}^L$ are sequences of aggregation and combine functions, $\{\sigma^{(i)}\}_{i=1}^L$ is a sequence of activation functions and CLS is a classification function.

Given a labeled graph $\mathfrak{G} = (V, E, g_V)$, the AC-GNN model computes vectors $\overline{x}_v^{(i)}$ for every $v \in V$ in every layer $1 \leq i \leq L$ as follows: $\overline{x}_v^{(0)} = g_V(v)$ is the initial feature vector of $v$, and for $i > 1$

$$\overline{x}_v^{(i)} = \sigma^{(i)}\left(\text{COM}^{(i)}\left(\overline{x}_v^{(i-1)}, \overline{y}\right)\right), \text{ where } \overline{y} = \text{AGG}^{(i)}\left(\{\!\!\{\overline{x}_u^{(i-1)} \mid u \in N_{\mathfrak{G}}(v)\}\!\!\}\right).$$

The classification function $\text{CLS}: \mathbb{R}^k \to \{0, 1\}$ is applied to the resulting feature vectors $\overline{x}_v^{(L)}$.

In this paper we focus on the real-valued computation part of GNNs and discard the classification function. We consider the feature vectors $\overline{x}_v^{(L)}$ after the computation of layer $L$ as our output. While we could also integrate CLS into our model, for our concerns this is not needed.

Next we define arithmetic circuits as a model of computation for computing real functions. Since we will put them in the context of graph neural networks, which inherently operate on vectors of real numbers rather than individual reals, we will accordingly define arithmetic circuits relative to $\mathbb{R}^k$ rather than relative to $\mathbb{R}$, as is done more commonly (cf. e.g. [Blum et al., 1997]).

**Definition 2.3.** Let $k, n, m \in \mathbb{N}$. An $\mathbb{R}^k$-*arithmetic circuit* with $n$ inputs and $m$ outputs is a simple directed acyclic graph of labeled nodes, also called *gates*, such that

- there are exactly $n$ gates labeled *input*, which each have indegree 0,
- there are exactly $m$ gates labeled *output*, which have indegree 1 and outdegree 0,
- there are gates labeled *constant*, which have indegree 0 and are labeled with a tuple $c \in \mathbb{R}^k$,
- there are gates labeled *projection*$_{i,j}$ for $1 \leq i, j \leq k$, which have in- and outdegree 1,
- there are gates labeled *addition* and *multiplication*.

Additionally, both the input and the output gates are ordered.

An $\mathbb{R}^k$-arithmetic circuit $C$ with $n$ inputs and $m$ outputs computes the function $f_C \colon (\mathbb{R}^k)^n \to (\mathbb{R}^k)^m$ as follows: Initially, the input to the circuit is placed in the input gates. Afterwards, in each step, each arithmetic gate whose predecessor gates all have a value, computes the respective function it is labeled with, using the values of its predecessors as inputs. By addition and multiplication we refer to the respective componentwise operations and the projection computes the function $proj_{i,j} \colon \mathbb{R}^k \to \mathbb{R}^k, \ (x_1, \ldots, x_i, \ldots x_k) \mapsto (0, \ldots, 0, \underset{\text{position } j}{x_i}, 0, \ldots, 0)$. Analogously, each output gate takes the value of its predecessor, once its predecessor has one. The output of $f_C$ is then the tuple of values in the $m$ output gates of $C$ after the computation.

The *depth* of $C$ (written *depth*$(C)$) is the length of the longest path from an input gate to an output gate in $C$ and the *size* of $C$ (written *size*$(C)$) is the number of gates in $C$. For a gate $g$ in $C$, we will write *depth*$(g)$ to denote the length of the longest path from an input gate to $g$ in $C$.

*Remark* 2.4. In the context of the circuits over $\mathbb{R}^k$ that we have just defined, a circuit over $\mathbb{R}$ is just a circuit over $\mathbb{R}^1$. Note that for circuits over $\mathbb{R}^1$, projection gates are just identity gates and can therefore be omitted. It should also be noted that for any fixed $k \in \mathbb{N}$, circuits over $\mathbb{R}^k$ and circuits over $\mathbb{R}$ can easily simulate each other. For a detailed explanation see the arXiv version Barlag et al. [2024].

Since an arithmetic circuit itself can only compute a function with a fixed number of arguments, we extend this definition to families of arithmetic circuits in a natural way.

**Definition 2.5.** An $\mathbb{R}^k$-*arithmetic circuit family* $\mathcal{C}$ is a sequence $(C_n)_{n \in \mathbb{N}}$ of circuits, where each circuit $C_n$ has exactly $n$ input gates. Its *depth* and *size* are functions mapping natural numbers $n$ to *depth*$(C_n)$ and *size*$(C_n)$, respectively.

An arithmetic circuit family $\mathcal{C} = (C_n)_{n \in \mathbb{N}}$ computes the function $f_{\mathcal{C}} \colon (\mathbb{R}^k)^* \to (\mathbb{R}^k)^*$ defined as $f_{\mathcal{C}}(\overline{x}) \coloneqq f_{C_{|\overline{x}|}}(\overline{x})$.

*Remark* 2.6. Since a circuit family is an infinite sequence of circuits, one would be hard pressed to consider such a family a (finite) algorithm. Such models of computation, where a different instance is needed for any different input length, is called *non-uniform*. In order to deem a circuit family $(C_n)_{n \in \mathbb{N}}$ to be an algorithm, a frequent requirement is the existence of an algorithm which, when given $n$ as an input, outputs the circuit $C_n$. Circuit families, for which such an algorithm exists are called *uniform* circuit families. For more details on circuit uniformity, see e.g. Vollmer [1999] and for more on uniformity with respect to real computation, see Blum et al. [1997].

**Definition 2.7.** For any $k \in \mathbb{N}$ and any two functions $s, d \colon \mathbb{N} \to \mathbb{N}$, FSIZE-DEPTH$_{\mathbb{R}^k}(s, d)$ is the class of all functions $(\mathbb{R}^k)^* \to (\mathbb{R}^k)^*$ that are computable by arithmetic circuit families of size in $\mathcal{O}(s)$ and depth in $\mathcal{O}(d)$. Let $\mathcal{A}$ be a set of functions of the form $f \colon \mathbb{R}^k \to \mathbb{R}^k$. Then FSIZE-DEPTH$_{\mathbb{R}^k}(s, d)[\mathcal{A}]$ is the class of functions computable by arithmetic circuits with the same constraints, but with additional gate types $g_f$, with indegree and outdegree 1 that compute $f$, for each $f \in \mathcal{A}$.

We call all classes of the form FSIZE-DEPTH$_{\mathbb{R}^k}(s, d)[\mathcal{A}]$ (and their subclasses) *circuit function classes*. As usual in circuit complexity, an $\mathfrak{F}$-*circuit family*, where $\mathfrak{F}$ is a class of functions, will denote a circuit family that computes a function in $\mathfrak{F}$.

We restrict this work to versions of one specific circuit function class.

**Definition 2.8.** FAC$^0_{\mathbb{R}^k}$ is the class of all functions $f \colon (\mathbb{R}^k)^* \to (\mathbb{R}^k)^*$ that can be computed by arithmetic circuit families $(C_n)_{n \in \mathbb{N}}$ of constant depth and polynomial size, i.e., FAC$^0_{\mathbb{R}^k} =$ FSIZE-DEPTH$_{\mathbb{R}^k}(n^{\mathcal{O}(1)}, 1)$.

By Definition 2.7, FAC$^0_{\mathbb{R}^k}[\mathcal{A}]$-circuit families are FAC$^0_{\mathbb{R}^k}$-circuit families where the circuits are expanded by additional gate types for all functions in $\mathcal{A}$.

A node in a graph neural network aggregates the values of its neighbors in each step, irrespective of their order, and then combines them with its own previous value. In order for our circuits to mimic this behaviour, we impose a symmetry condition on them. We require the functions they compute to be *tail-symmetric*, i.e., to only be able to single out their first argument.

**Definition 2.9.** A function $f\colon (\mathbb{R}^k)^* \rightarrow (\mathbb{R}^k)^*$ is *tail-symmetric*, if $f(\overline{x}_1, \ldots, \overline{x}_n) = f(\overline{x}_1, \pi(\overline{x}_2, \ldots, \overline{x}_n))$ for all permutations $\pi$.

*Remark* 2.10. One way to construct a tail-symmetric function $f$ is to take a binary function $g$ and a fully symmetric function $h$ of arbitrary arity and compose them: $f(x_1, \ldots, x_n) := g(x_1, h(x_2, \ldots, x_n))$. Note that this is precisely the notion required for the aggregate-combine step in AC-GNNs as per Definition 2.2.

An arithmetic circuit $C$ is *tail-symmetric* if $f_C$ is tail-symmetric. In the sequel, we will consider families of tail-symmetric functions computed by some circuit family $\mathcal{C}$. In this setting, the circuit family computes one unique function $f_n$ for each arity $n \in \mathbb{N}$. We will then write $f_{\mathcal{C}}\left(\overline{x}_1, \{\!\{\overline{x}_2, \ldots, \overline{x}_n\}\!\}\right)$ to denote $f_n(\overline{x}_1, \ldots, \overline{x}_n)$, when $\{\!\{\overline{x}_2, \ldots, \overline{x}_n\}\!\}$ is a multiset of cardinality $n - 1$.

**Definition 2.11.** Let $\mathfrak{F}$ be a circuit function class. We denote by $t\mathfrak{F}$ the class which contains all functions of $\mathfrak{F}$ that are tail-symmetric.

# 3 Graph Neural Networks using Circuits

We now define the main computation model of interest for this paper, namely circuit graph neural networks (C-GNNs). C-GNNs work similarly to plain AC graph neural networks. Instead of using the concept of a two-step aggregate and combine computation via which the new feature vector of a node is computed, C-GNNs permit that this computation is done by an arithmetic circuit (with particular resource bounds). This allows us to classify GNNs based on the complexity of their internal computing functions.

## 3.1 Model of Computation

A *basis* of a C-GNN is a set of functions of a specific circuit function class along with a set of activation functions. The respective networks will be based on these functions.

**Definition 3.1.** Let $\mathcal{S}$ be a non-empty set of functions from $(\mathbb{R}^k)^*$ to $(\mathbb{R}^k)^*$ and let $\mathcal{A}$ be a non-empty set of activation functions of dimension $k$. Then we call the set $\mathcal{S} \times \mathcal{A}$ a *C-GNN-basis* of dimension $k$.

C-GNNs of a particular basis essentially consist of a fixed depth and a function assigning circuit families and activation functions from its basis to its different layers.

**Definition 3.2.** Let $B = \mathcal{S} \times \mathcal{A}$ be a C-GNN-basis of dimension $k$. A *circuit graph neural network (of dimension $k$)* of depth $d \in \mathbb{N}$ is a function $\mathcal{N}\colon [d] \to B$. If all functions in $\mathcal{S}$ belong to a circuit function class $\mathfrak{F}$, we also say that $\mathcal{N}$ is a $(\mathfrak{F}, \mathcal{A})$-GNN.

**Definition 3.3.** Let $\mathcal{N}$ be a C-GNN of depth $d$. The function $f_{\mathcal{N}}\colon \mathfrak{Graph}_k \to \mathfrak{Graph}_k$ computed by $\mathcal{N}$ is defined as follows:

Let $\mathfrak{G} = (V, E, g_V)$ be a labeled graph. The labeled graph $\mathfrak{G}'$ computed by $f_{\mathcal{N}}(\mathfrak{G})$ has the same structure as $\mathfrak{G}$, however, its nodes have different feature vectors. That is $f_{\mathcal{N}}(\mathfrak{G}) = \mathfrak{G}'$, where $\mathfrak{G}' = (V, E, h_V)$ and $h_V$ is defined inductively as follows:

$$h_V^{(0)}(w) := g_V(w)$$
$$h_V^{(i)}(w) := \sigma^{(i)}\left(f_{\mathcal{C}^{(i)}}\left(h_V^{(i-1)}(w), M\right)\right), \text{ with } M = \left\{\!\!\left\{ h_V^{(i-1)}(u) \mid u \in N_{\mathfrak{G}}(w) \right\}\!\!\right\}$$

where $\mathcal{N}(i) = \left(\mathcal{C}^{(i)}, \sigma^{(i)}\right)$. Finally $h_V(w) := h_V^{(d)}(w)$.

The C-GNN model that we have just defined computes functions from labeled graphs to labeled graphs (with the same graph structure).

Our results are stated for C-GNNs using functions from versions of the function class $\mathrm{FAC}^0_{\mathbb{R}^k}$ as defined in Definitions 2.8 and 2.11.

*Remark* 3.4. We can relate our C-GNN model to the continuous computation of traditional AC-GNNs. For any AC-GNN following Definition 2.2 with activation functions $\mathcal{A}$ and where the aggregation functions are computable by $\text{FAC}^0_{\mathbb{R}^k}$-circuit families, there exists a $(\text{FAC}^0_{\mathbb{R}^k}, \mathcal{A})$-GNN with a constant number of layers which computes the same function, omitting only the functionality of the classification function. This holds for the common aggregation functions like the sum, product or mean.

## 3.2 Simulating C-GNNs with arithmetic circuits

**Definition 3.5.** Let $\mathfrak{G} = (V, E, g_V)$ be a labeled graph, $n := |V|$, and $M = \text{adj}(\mathfrak{G})$ be the adjacency matrix of $(V, E)$, where the columns are ordered in accordance with the ordering of $V$. We write $\langle M \rangle$ to denote the encoding of $M$ as the $n^2$ matrix entries written using vectors $(0, \ldots, 0)$ and $(1, \ldots, 1)$ in $\mathbb{R}^k$, ordered in a row wise fashion. We denote them by $\overline{m}_{ij}$ encoding the matrix entry $m_{ij}$. We write $\langle \mathfrak{G} \rangle$ to denote the encoding of $\mathfrak{G}$ as a tuple of real valued vectors, such that $\langle \mathfrak{G} \rangle = (\langle M \rangle, \text{vec}(\mathfrak{G})) \in (\mathbb{R}^k)^{n^2+n}$, which consists of the encoding of $M$ followed by $\text{vec}(\mathfrak{G})$, the feature vectors of $\mathfrak{G}$. The feature vectors $\text{vec}(\mathfrak{G})$ are $g_V(v)$ for all $v \in V$ and ordered like $V$.

**Theorem 3.6.** *Let $\mathcal{N}$ be a $(t\text{FAC}^0_{\mathbb{R}^k}[\mathcal{A}], \{\text{id}\})$-GNN. Then there exists an $\text{FAC}^0_{\mathbb{R}^k}[\mathcal{A}]$-circuit family $\mathcal{C}$, such that for all labeled graphs $\mathfrak{G}$ the following holds: $f_{\mathcal{N}}(\mathfrak{G}) = \mathfrak{G}'$ iff $\mathcal{C}(\langle \mathfrak{G} \rangle) = \langle \mathfrak{G}' \rangle$, where $\langle \mathfrak{G} \rangle = (\langle \text{adj}(\mathfrak{G}) \rangle, \langle \text{vec}(\mathfrak{G}) \rangle)$.*

*Proof.* We essentially roll out the given $(t\text{FAC}^0_{\mathbb{R}^k}[\mathcal{A}], \{\text{id}\})$-GNN $\mathcal{N}$ by using a circuit to simulate each individual layer of $\mathcal{N}$ and concatenating these circuits to then simulate $\mathcal{N}$. Since the computation in each node of $\mathcal{N}$ can be performed by a circuit of an $\text{FAC}^0_{\mathbb{R}^k}[\mathcal{A}]$-family, simulating this computation can be done without an issue. Similarly, connecting the simulated nodes and layers can be done relatively simply, and thus the resulting circuit family is an $\text{FAC}^0_{\mathbb{R}^k}[\mathcal{A}]$-family. The full proof can be found in the arXiv version Barlag et al. [2024]. □

*Remark* 3.7. Note that this proof preserves uniformity: if the sequence of C-GNNs is uniform, so is the circuit family. Furthermore, for any set of activation functions $\mathcal{A}$, $(t\text{FAC}^0_{\mathbb{R}^k}[\mathcal{A}], \{\text{id}\})$-GNNs are at least as powerful as $(t\text{FAC}^0_{\mathbb{R}^k}, \{\text{id}\} \cup \mathcal{A})$-GNNs and $(t\text{FAC}^0_{\mathbb{R}^k}, \{\text{id}\})$-GNNs. The latter do not have access to the functions of $\mathcal{A}$ at all and $(t\text{FAC}^0_{\mathbb{R}^k}, \{\text{id}\} \cup \mathcal{A})$-GNNs are more restricted in the use of the functions than $(t\text{FAC}^0_{\mathbb{R}^k}[\mathcal{A}], \{\text{id}\})$-GNNs.

## 3.3 Simulating arithmetic circuits with C-GNNs

In order to simulate $\text{FAC}^0_{\mathbb{R}^k}$-circuit families using C-GNNs, we utilize the following normal form.

**Definition 3.8.** A circuit $C$ is in *path-length normal form* if every path from an input to an output gate has the same length and every non-input and non-output gate has exactly one successor.

**Lemma 3.9.** *Let $\mathcal{C}$ be an $\text{FAC}^0_{\mathbb{R}^k}$-circuit family. Then for every circuit $C \in \mathcal{C}$ there exists a circuit $C'$ of the same depth in path-length normal form such that $f_{C'}(\overline{x}) = f_C(\overline{x})$, for all $\overline{x} \in (\mathbb{R}^k)^*$.*

*Proof.* We describe a procedure that transforms $C$ to $C'$. If a gate $h$ in $C$ has more than one successor, the subgraph consisting of $h$ and all paths from input gates to $h$ are copied in $C'$ for each successor of $h$. These copies are then used as the respective input for those successors which results in every gate in $C'$ having only one successor. This is done iteratively, where in each step only gates in one depth layer of the circuit are modified, starting with depth 1, the gates closest to the input gates. Each step in this procedure incurs a polynomial overhead in size. Since $C$ is of constant depth there are only constantly many iteration steps and thus $C'$ remains polynomial in size.

Let $\ell_{max} = depth(C)(= depth(C'))$ be the length of the longest path from an input to an output node in $C$. For every path of length $\ell$ which is shorter than $\ell_{max}$, we add to $C'$ $\ell_{max} - \ell$ unary addition gates directly after the input. The depth of the resulting circuit $C'$ equals the one of $C$, as all changes to path lengths are made with this upper bound in mind. Since all changes made preserve the computed function and the produced $C'$ is in path-length normal form, the claim follows. □

Since C-GNNs get labeled graphs as their input, while arithmetic circuits typically work on vectors over $\mathbb{R}^k$, we show next how to represent a circuit as an input to a C-GNN.

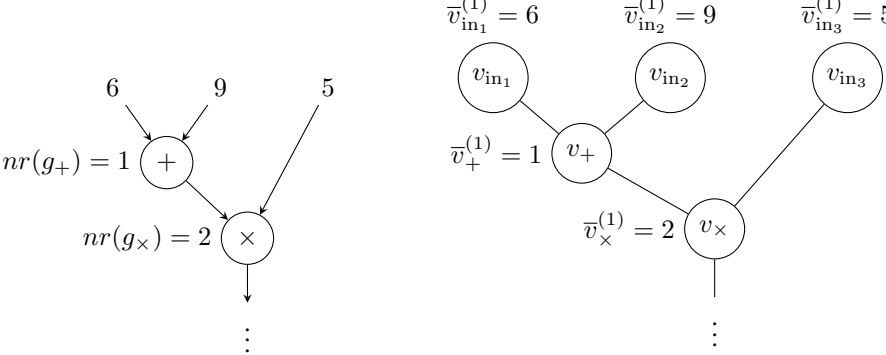

(a) Underlying circuit, to be simulated by the C-GNN ($nr(*)$ denotes the unique number of the gate).

(b) The labeled graph and initial feature vectors of the C-GNN ($v_g^{(i)}$ denotes the feature vector of a gate at layer $i$).

Figure 1: Example illustrating the proof of Theorem 3.11. [3]

Table 1: Example illustrating the proof of Theorem 3.11: The values of the feature vectors during the computation of the C-GNN.

| LAYER | $\overline{v}_{in_1}$ | $\overline{v}_{in_2}$ | $\overline{v}_{in_3}$ | $\overline{v}_+$ | $\overline{v}_\times$ |
|---|---|---|---|---|---|
| 1 | 6 | 9 | 5 | 1 | 2 |
| 2 | 6 | 9 | 5 | $6 + 9 = 15$ | 2 |
| 3 | 6 | 9 | 5 | 15 | $5 \times 15 = 75$ |

**Definition 3.10.** If $C$ is a circuit with input $\overline{x}$ whose gates are additionally numbered with unique elements of $\mathbb{R}$, let $\mathfrak{G}_{C,\overline{x}}$ be the labeled graph $(G, g_V)$, where $G = (V, E)$ is the underlying graph of the circuit and the function $g_V$ is defined as follows. The feature vector of a graph node corresponding to an input gate is the corresponding input value. Each other node gets its own number according to the gate numbering of the circuit as its feature vector.

The following theorems describe how C-GNNs are able to simulate arithmetic circuits. The proofs are given for $\mathbb{R}$ circuits, but as mentioned in Remark 2.4 families of constant depth and polynomial size $\mathbb{R}^k$ and $\mathbb{R}$ circuits can simulate each other with constant increase/decrease in size, so they also hold for $\mathbb{R}^k$ circuits.

**Theorem 3.11.** *Let* $\mathcal{C} = (C_n)_{n \in \mathbb{N}}$ *be an* $\mathrm{FAC}^0_{\mathbb{R}}$*-circuit family. Then there exists a* $(t\mathrm{FAC}^0_{\mathbb{R}}, \{\mathrm{id}\})$*-GNN* $\mathcal{N}$*, such that for all* $n \in \mathbb{N}$ *and* $\overline{x} \in \mathbb{R}^n$*, the feature vectors of the nodes of* $f_{\mathcal{N}}(\mathfrak{G}_{C_n,\overline{x}})$ *corresponding to output gates in* $C_n$ *are exactly the components of* $f_{C_n}(\overline{x})$ *in the same order. The number of layers in* $\mathcal{N}$ *is equal to the depth of* $\mathcal{C}$*.*

*Proof.* The general idea is to use the graphs of the circuits from $\mathcal{C}$ as input graphs for a C-GNN and then simulate the gates layerwise. The gates of $\mathcal{C}$ are numbered and the numbering is represented in the feature vectors for each node. In each layer circuits of the C-GNN use a lookup table to decide which operation is applied to a feature vector. An example illustrating the proof idea is given in Figure 1 and Table 1. For simplicity reasons an example omitting any special cases is chosen. The full proof can be found in the arXiv version Barlag et al. [2024]. $\square$

Since we have shown in Section 3.2 that C-GNNs can be simulated by $\mathrm{FAC}^0_{\mathbb{R}^k}[\mathcal{A}]$-circuit families we now examine the converse direction: simulating these families with C-GNNs.

**Theorem 3.12.** *Let* $\mathcal{C} = (C_n)_{n \in \mathbb{N}}$ *be an* $\mathrm{FAC}^0_{\mathbb{R}}[\mathcal{A}]$*-circuit family. Then there exists a* $(t\mathrm{FAC}^0_{\mathbb{R}}[\mathcal{A}], \{\mathrm{id}\})$*-GNN* $\mathcal{N}$*, such that for all* $n \in \mathbb{N}$ *and* $\overline{x} \in \mathbb{R}^n$*, the feature vectors of the nodes of* $f_{\mathcal{N}}(\mathfrak{G}_{C_n,\overline{x}})$ *corresponding to output gates in* $C_n$ *are exactly the components of* $f_{C_n}(\overline{x})$ *in the same order. The number of layers in* $\mathcal{N}$ *is equal to the depth of* $\mathcal{C}$*.*

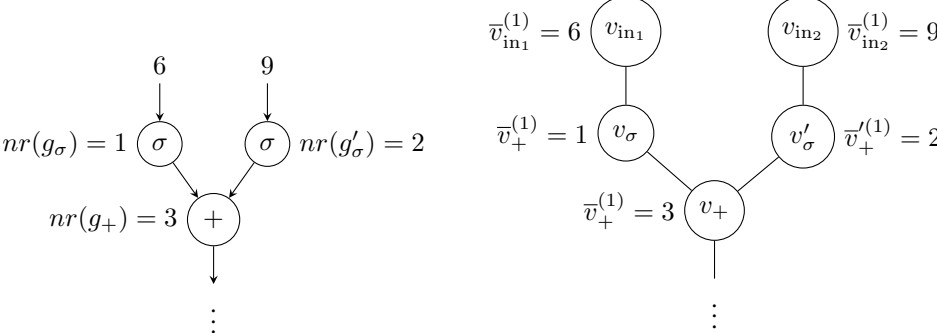

(a) Underlying circuit, to be simulated by the C-GNN ($nr(*)$ denotes the unique number of the gate).

(b) The labeled graph and initial feature vectors of the C-GNN ($v_g^{(i)}$ denotes the feature vector of a gate at layer $i$).

Figure 2: Example illustrating the proof of Theorem 3.17. [3]

Table 2: Example illustrating the proof of Theorem 3.17: The values of the feature vectors during the computation of the C-GNN.

| LAYER | $\overline{v}_{in_1}$ | $\overline{v}_{in_2}$ | $\overline{v}_\sigma$ | $\overline{v}'_\sigma$ | $\overline{v}_+$ |
|---|---|---|---|---|---|
| 1 | 6 | 9 | 1 | 2 | 3 |
| 2 | $\sigma(6)$ | $\sigma(9)$ | $\sigma(1)$ | $\sigma(9)$ | $\sigma(\sigma^{-1}(3)) = 3$ |
| 3 | $\sigma(6)$ | $\sigma(9)$ | $\sigma(1)$ | $\sigma(9)$ | $\sigma(6) + \sigma(9)$ |

*Proof.* The proof follows the same concept as the proof of Theorem 3.11. With the exception that the internal circuits of the C-GNN now have access to the same set of functions $\mathcal{A}$ in the form of gates that compute them. Those are then used to simulate gates from $\mathcal{C}$ that compute functions of $\mathcal{A}$. The full proof can be found in the arXiv version Barlag et al. [2024]. □

In the previous two theorems, we have shown that C-GNNs can simulate $FAC^0_\mathbb{R}$-circuit families as is and that they can simulate $FAC^0_\mathbb{R}[\mathcal{A}]$-circuit families if they have arbitrary access to the functions in $\mathcal{A}$. If we consider $\mathcal{A}$ to consist of activation functions, though, it would be more natural to only permit them as such in our C-GNN model. In the following we will introduce a structural restriction of circuits, which will allow us to capture a subset of $FAC^0_\mathbb{R}[\mathcal{A}]$ using C-GNNs that have access to $\mathcal{A}$ only as activation functions.

**Definition 3.13.** A circuit $C$ is said to be in *function-layer form* if it is in path-length normal form and for each depth $d \leq depth(C)$ all gates of $C$ at depth $d$ have the same gate type.

**Definition 3.14.** Let $\mathfrak{F}$ be a circuit function class. We denote by $f\mathfrak{F}$ the class which contains all functions of $\mathfrak{F}$ that can be computed by circuit families where all circuits are in function-layer form.

We also need a small restriction on the functions we will permit as activation functions.

**Definition 3.15.** A computable function $f: \mathbb{R}^k \to \mathbb{R}^k$ is *countably injective* if there is a countably infinite set $S \subseteq \mathbb{R}^k$ such that $f{\restriction}_S$ is injective. We also say that $f$ is countably injective on $S$.

*Remark* 3.16. The commonly used activation functions $\text{ReLU}(x) = \max(0, x)$, $\sigma(x) = \frac{1}{1+e^{-x}}$ and $\tanh(x) = \frac{e^x - e^{-x}}{e^x + e^{-x}}$ are countably injective.

**Theorem 3.17.** *Let $\mathcal{A}$ be a set of countably injective activation functions and let $\mathcal{C} = (C_n)_{n \in \mathbb{N}}$ be a $fFAC^0_\mathbb{R}[\mathcal{A}]$-circuit family. Then there exists a $(tFAC^0_\mathbb{R}, \mathcal{A} \cup \{id\})$-GNN $\mathcal{N}$, such that for all $n \in \mathbb{N}$ and $\overline{x} \in \mathbb{R}^n$, the feature vectors of the nodes of $f_\mathcal{N}(\mathfrak{G}_{C_n,\overline{x}})$ corresponding to output gates in $C_n$ are exactly the components of $f_{C_n}(\overline{x})$ in the same order. The number of layers in $\mathcal{N}$ is equal to the depth of $\mathcal{C}$.*

*Proof.* Contrary to the proof of Theorem 3.12 the gates in $\mathcal{C}$ computing a function $\sigma \in \mathcal{A}$ are simulated in the C-GNN by using the respective function as an activation function applied to all

nodes. To distinguish between nodes where this function should and should not be applied we assign $\sigma^{-1}$ of the respective feature vectors of nodes where we do not want to change the value. This is done prior to the application of the activation function. Therefore the function needs to be injective on some countable set. An example illustrating the proof idea is given in Figure 2 and Table 2. As in the proof of Theorem 3.17 an example omitting any special cases is chosen. The full proof can be found in the arXiv version Barlag et al. [2024]. □

## 4 Conclusion

In this paper we showed a correspondence between arithmetic circuits and a generalization of graph neural networks using circuits. However, some restrictions needed to be imposed on the particular circuits used in our constructions. In particular, the circuits used in our C-GNNs need to be tail symmetric, and in Theorem 3.17 the corresponding $\text{FAC}^0_{\mathbb{R}}[\mathcal{A}]$-circuit family needs to be in function-layer form as well. An interesting avenue for further research is the question, whether those restrictions can be made "on both sides in our simulations". This means, in particular, to ask if $(t\text{FAC}^0_{\mathbb{R}^k}, \{id\} \cup \mathcal{A})$-GNNs can be simulated by $f\text{FAC}^0_{\mathbb{R}}[\mathcal{A}]$ families and, if $(tf\text{FAC}^0_{\mathbb{R}^k}, \{id\} \cup \mathcal{A})$-GNNs and $f\text{FAC}^0_{\mathbb{R}^k}[\mathcal{A}]$ can simulate each other. Another direction of further work is to study whether by relating our C-GNNs to variants of so-called VVc-GNNs of Sato et al. [2019], we may drop the assumption of tail-symmetry.

We already mentioned the issue of uniformity. If the sequence of GNNs, one for each graph size, is uniform in the sense that there is an algorithm that, given the size of the graph as an input, outputs the GNN responsible for that size, then the sequence of arithmetic circuits will also be uniform, because our simulation proof explicitly shows how to construct the circuit; and vice versa. In future work, this should be made more precise. In the case of Boolean circuits, logtime-uniformity ($\text{U}_{E*}$-uniformity) has become the standard requirement [Vollmer, 1999]. What is the corresponding precise uniformity notion for GNNs?

In the introduction we mentioned the complexity of the problem of training neural networks. We studied the expressiveness (or computational power) of neural networks. The higher the expressiveness of a network is, the more complicated the training process will be; however, the question to decide if a trained network of given quality exists might become easier to decide. Can this be made precise? Is there a formal result connecting complexity of training and network expressiveness?

Further investigations are required to obtain practical implications of our theoretical results. Once we fix the architecture of our C-GNNs (i.e., a circuit function class for the GNN nodes), we obtain a specific circuit class that characterizes the computational power of the C-GNN model which can be then rigorously studied. The question of what can be computed by arithmetic circuits using activation functions is related to the question of which functions can be computed using other functions, or in other words, which functions are "more complex" than others. We are only aware of very little work in this area. Results include a PTIME upper bound (in the so-called *BSS*-model of computation) for the arithmetic circuit complexity class $\text{NC}_{\mathbb{R}}$ that uses the sign function Cucker [1992].

As it currently stands, there does not seem to be meaningful experiments that could be run for our model of computation. The problem is that not much is known about the computational power of the circuit classes we utilize. Further research into functions computable by practically implementable arithmetic circuits of different sizes and depths would be required. This could e.g. be done by investigating the Boolean parts of different complexity classes defined by families of arithmetic circuits, i.e., the classes when restricted to Boolean inputs. This research would pinpoint properties that would be provably outside of the capabilities of particular GNN models, whose learnability could be then tested in practice. We believe that our results will motivate research for these circuit classes.

Merrill et al. [2022], Merrill and Sabharwal [2022] have studied transformer networks from a computational complexity perspective and obtained a $\text{TC}^0$ upper bound, similar to the result for GNNs by Grohe [2023]. We think it is worthwhile to study the computational power of transformers from the point of view of computation over the reals, similar to what we have done in this paper for GNNs.

---

[3]For readability purposes the one-dimensional vectors in the C-GNN are written as real numbers.

## Acknowledgements

The first and the fourth author were partially supported by the DFG grant VI 1045-1/1.

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
