# OpenReview forum: "Graph Neural Networks and Arithmetic Circuits"
_NeurIPS.cc/2024/Conference — NeurIPS 2024 poster_

### Official Review · Reviewer_2Yjw · 2024-07-02

**Soundness:** 4
**Presentation:** 2
**Contribution:** 3
**Rating:** 7
**Confidence:** 3

**Summary:**

The paper establishes a series of expressiveness/expressive efficiency equivalence between graph neural networks (GNNs) whose aggregation/combine operations can be computed by arithmetic circuits, and various sub-classes of constant-depth arithmetic circuits. The main challenge consists of showing expressiveness equivalences in the case of the GNN or the aggregation/combine operations being equipped with some non-linearities. In principle, one could then leverage these expressive equivalence results to show limitations of what GNNs can compute through the language of arithmetic circuits.

**Strengths:**

I believe the paper brings a substantial theoretical contribution to the problem of understanding the expressiveness of GNNs, since it proposes using a special class of arithmetic circuits to express the computations of GNNs (and vice versa). By doing so, any limitation shown over this class of arithmetic circuits can be translated to GNNs as well.

The proofs of the theorems are briefly discussed in the main text, and then they are fully presented in the appendix. I believe the proofs in the main text do quite a good job in giving an outline about the full proof. Most of the notation and terminology is very well-explained formally, even though in just a few cases they are not (see weaknesses).

**Weaknesses:**

While I do not have a background about showing expressiveness results of GNNs, I have some background in doing so for arithmetic circuits. The great flexibility of arithmetic circuits when equipped with non-linearities (as the ones showed in the paper) suggests it might be very difficult showing their limitations, and therefore limitations for the GNNs in this way. Therefore, I think the class of arithmetic circuits considered might limit the impact of the contribution to being important but not exceptional. I am happy to reconsider my score if proved otherwise (see also my questions).

I think there are some weaknesses regarding the presentation at multiple points in the paper (hence the 2/4 score), which I list below:
- The introduction is too long (~3 pages) and I was often confused about which content in it was actually relevant for the paper. For example, the first paragraph (L9-16) is not relevant, as also mentioned in L17. Also, L26-50 discuss relationships with fragments of first-order logics in the literature, but this sounds far from the actual contributions of the paper. I have noticed there is no related work section in the paper. I would suggest moving some content of the introduction to a related work section just before the conclusion, if it is really something related to the paper. This would make the contributions appear sooner.
- The contributions of the paper are listed at multiple points of the introduction, and some concepts are often repeated. For instance, L68-72, L106-109, L114-116 look all similar sentences. I think it is sufficient to list the contribution and motivate them (e.g., L129-134) only once.
- The authors should check if all concepts are explained in the introduction before using them, such as "existential theory of real numbers" and the class $\mathrm{FAC}_R^0$ (L86).
- I was quite confused about the meaning of L253, since projection is used also as a gate in the arithmetic circuit. Here it is instead using to extract one element of the pair returned by the circuit GNN. Also, I was expecting $\sigma^{(i)}$ to be a non-linearity (e.g., ReLU), but it is instead the projection here. I think L252-L254 can be rephrased as to make the notation more clear.
- L202 uses the star symbol. I suppose this is the Kleene star operator, but it seems to not be defined.
- Definition 3.15 contains an upward half arrow labelled with $S$. What is the semantic of this symbol? Is this defined somewhere? If not, it is better to define it.
- Since there is a high number of definitions appearing way before the theorems using them, I was wondering if you could remove some definitions. For example, do you need $\mathrm{FSIZE-DEPTH}_{\mathbb{R}^k}(s,d)[\mathcal{A}]$  as a concept in other parts of the paper? Isn't it sufficient to just introduce Definition 2.7 without using it? It looks like the paper only deals with constant-depth circuits anyway.

I strongly recommend all these issues to be eventually fixed in the camera ready of the paper, which should require modest effort.

**Questions:**

I list my questions below:
1. It seems the paper does not show a theoretical application of the expressive equivalences shown. Do the authors have some insight about which kind of limitations of GNNs can be shown through the language of arithmetic circuits? The arithmetic circuits equipped with non-linear gates appears quite powerful. Are there already some works on showing limitations of what they can compute?
2. The authors introduce the concept of arithmetic circuits computing tail-symmetric functions. What would be a systematic way to build tail-symmetric circuits in practice?
3. The proof of Lemma 3.9 shows how to decompose an arithmetic circuit whose maximum out-degree of nodes is > 1 into another one whose out-degree of nodes is instead 1. Do the authors agree that the space complexity of such decomposition is polynomial in the size of the original circuit only because we are considering constant-depth circuits? If the circuits were not constant-depth, then the complexity of the decomposition would be exponential w.r.t. the depth, right? If this is true, then the authors should make this more clear in the proof.
4. It is not clear to me at which point of the proof of Theorem 3.17 we need the activation functions to be countably injective. Would it be fine if they were simply bijective? Can you please explain why do we need this constraint?
5. The authors mention multiple times that their results are "uniform", but it seems it is not properly defined anywhere. What is a formal definition of "uniformity" in general? Why is it important?

**Limitations:**

The only limitation I can think of is related to the theoretical results is that in many cases they require using arithmetic circuits equipped with non-linear gates, for which showing limitations in what they can compute sounds quite hard. See my question 1 above for details.

---

> ### Author Rebuttal · Authors · 2024-08-06
>
> ## Weaknesses
>
> 1\., 2\. & 3\. We will carefully revise our introduction to eliminate repetition and to ensure that all concepts are sufficiently treated. Currently our introduction starts with a subsection "Background and Related Work", we will split this section into two parts and collect relevant related work discussion to a separate "Related Work" subsection to the end of the introduction. We strongly believe that our work should be put into its context in the introduction by discussing related work there in a sufficient manner.
>
> 4\. & 5\. Thanks for pointing this out. The star is indeed the Kleene star and the upward half arrow defines the restriction of a function. We will address this in the paper.
>
> 6\. We agree that there is high number of definitions, but our aim is to have a general framework to be able to reason about the complexity of GNNs. We therefore defined the C-GNNs to be as general as possible not just restricting ourselves to a version with only circuits of constant depth.
> ## Questions
> 1. Once we fix the circuit and the activation function we get the connection to the specific circuit class this circuit belongs to and are able to study it. Concerning the limits of what can be computed by arithmetic circuits using activation functions, we would like to mention that this is related to the question of which functions can be computed using other functions, or in other words, which functions are "more complex" than others. We are only aware of very little work in this area. Results include  an upper bound for arithmetic circuit complexity classes with the addition of the sign function shown by Cucker (Felipe Cucker. PR != NCR . J. Complexity, 8(3):230–238, 1992]).
> 2. We are not entirely sure if we understand this question correctly. Are you asking if there exists a syntactic characterization of the notion of tail-symmetry, since this notion is purely semantical? If so, this is indeed a very good question that would be interesting to look into in the future.
> 3. This is indeed true, thanks for pointing this out, we will make this more clear in our proof.
> 4. We need the countable injectivity in activation functions for simulating circuits by using C-GNNs, since we encode the gate numbers of the circuit we are trying to simulate in the preimage of our activation functions. This way, we retrieve the gate number whenever we apply the activation function at the end of the computation of a node in our C-GNN. In general since countable injectivitiy is the weaker requirement bijectivity would also suffice.
> 5. Any fixed arithmetic circuit has a fixed number of inputs. This means that any circuit can only work on inputs of a fixed length. This is why we defined families of such circuits for our results. Computational models with this property, i.e., where essentially a "different machine" is needed for every input length are generally called non-uniform. If one can by computational means obtain the circuits of a given family, this family is then called uniform, as there exists an algorithm to compute on every input length. In Boolean complexity theory, oftentimes a Turing machine outputting the nth circuit on input n is used as a witness of a circuit family's uniformity.
> Uniformity is important, as it provides us with an algorithm that can work on inputs of arbitrary length, whereas the existence of e.g. a nonuniform circuit family that computes a particular function cannot really be seen as an implementable algorithm.
> From the constructive nature of our proofs, it follows that we will have some notion of uniformity, since they essentially define an algorithm to obtain the nth circuit or C-GNN. Defining that notion precisely will still require some work.

---

> ### Comment · Reviewer_2Yjw · 2024-08-08
>
> >Are you asking if there exists a syntactic characterization of the notion of tail-symmetry, since this notion is purely semantical?
>
> Yes, or better, I am asking what are the possible ways to construct tail-symmetric functions. Can you also provide some examples?
>
> >Defining that notion precisely will still require some work.
>
> Thank you for the very clear explanation.
> I still suggest the authors to try to give an outline of what "uniform" means in the paper.
>
>
> I have updated my score as to reflect the answers given by the authors.

---

> > ### Author Response · Authors · 2024-08-12
> >
> > *Regarding tail symmetric functions:* One very simple way of constructing tail-symmetric functions would be by defining a circuit that computes a symmetric function in arbitrarily many inputs $f(x_1, ..., x_n)$ (e.g. the sum of all inputs) and one that computes a binary function $g(x_1, x_2)$ (e.g. a linear function in two variables). If one then takes the result of the symmetric function as the second input to the binary function, i.e., $h(x_1, ..., x_n) := g(x_1, f(x_2, ..., x_(n+1)))$, then the resulting functin $h$ is tail-symmetric. This particular example would yield one way to define aggregate-combine GNNs as per Barceló et al. [Pablo Barceló, Egor V. Kostylev, Mikael Monet, Jorge Pérez, Juan Reutter, Juan Pablo Silva. The Logical Expressiveness of Graph Neural Networks. ICLR 2020.], but of course any kind of such functions are possible.
> > It should be noted, that this is by no means a complete characterization of such functions and that investigating functions of that kind would be an interesting avenue for further research. In particular, we deem it interesting to study circuits which compute tail-symmetric functions and ideally obtain a normal form of such circuits, i.e., a syntactical characterization for all functions of that kind.
> >
> > *Regarding uniformity:* We will add some comments about uniformity to the main body of the paper.

---

### Official Review · Reviewer_FVmx · 2024-07-08

**Soundness:** 3
**Presentation:** 3
**Contribution:** 3
**Rating:** 6
**Confidence:** 3

**Summary:**

The paper investigates the computational power of GNNs by demonstrating that the expressiveness of GNNs with different activation functions is equivalent to the capabilities of arithmetic circuits over real numbers. The authors introduce a new GNN variant called C-GNNs, which are equipped with constant-depth arithmetic circuits, and prove that these can compute the same functions as constant-depth arithmetic circuits. This result holds uniformly for all common activation functions and provides insights into the inherent computational limitations of GNNs, suggesting that enhancing GNN expressivity requires more complex functions or different architectures beyond mere scaling.

**Strengths:**

The paper offers a novel perspective on the expressiveness of GNNs, which is a valuable addition to the existing body of research on GNN expressiveness. This could potentially inspire future work in GNN design.

**Weaknesses:**

1. The paper's presentation may not align well with the expectations of a NeurIPS audience. Even for readers with a background in TCS and combinatorics, the contributions of the paper could be made more apparent. An improved presentation could potentially result in a higher evaluation score. (Questions 1, 2, 3)

2. The paper would benefit from a more thorough discussion and comparison with existing literature within the main body of the text. This would help to contextualize the paper's contributions within the broader field. (Questions 3, 4)

**Questions:**

1. In Line 115-116, you mention that the results are "not only an upper bound for neural networks." However, existing studies often establish the *equivalence* between GNNs and classic algorithms, implying both upper and lower bounds. Moreover, the exact correspondence seems to be established only for C-GNNs and not AC-GNNs. Could you clarify this distinction?

2. You state that the "result holds for all commonly used activation functions." Yet, the results appear to hinge solely on the injectivity of the activation function, without more detailed results. Since commonly adopted activation functions are injective, it's challenging to discern the expressiveness of these functions. Could you provide more nuanced insights into this matter?

3. The paper does not discuss applications to deep learning, which may limit its appeal to the NeurIPS audience.
- Could you elaborate on potential practical problems that the FAC-circuit family can solve or not solve?
- Additionally, how do FAC-circuits relate to traditional GNN expressiveness results, such as the WL test and first-order logic? Can these classic algorithms be simulated by FAC-circuits?

4. An important related work is missed [1]. This study discusses the equivalence between GNNs and a family of distributed algorithms, as well as the impact on aggregation functions. Furthermore, VVc-GNN appears similar to the C-GNN proposed in your paper. A discussion on this would be beneficial.

[1] Sato, Ryoma, Makoto Yamada, and Hisashi Kashima. "Approximation ratios of graph neural networks for combinatorial problems." NeurIPS 2019.

**Limitations:**

Yes

---

> ### Author Rebuttal · Authors · 2024-08-06
>
> **Regarding the mentioned weaknesses:**
> Thank you for pointing out these issues, we will definitely keep these in mind when revising the paper. We strife to appeal to the general NeurIPS audience, and for this purpose focussed also to presenting a thorough general introduction to the paper. This is though a bit of a balancing act given the theoretic nature of the result.
>
> **Regarding the questions:**
> 1. That is certainly correct. Our intention was not to imply that there aren't other precise characterisation results, instead our intent was to emphasize that our results produces a matching lower and upper bound. Furthermore, the direct correspondence we establish is indeed only achieved for C-GNNs. For AC-GNNs, we merely show an upper bound.
> 2. We wanted to emphasize that our results cover all commonly used activation functions. The proof technique does rely only on the injectivity of the activation function, and therefore holds for many more functions than just the ones mentioned. The main take away from this should be that if an activation function satisfies a very easy requirement (injectivity), then we get a correspondence between circuits and C-GNNs that have access to the same activation functions.
> 3. Questions about what different FAC-families can and cannot compute in practice still require further research. We believe that this work produces an application motivating the research on this topic for people working in circuit complexity. Since we are only concerned with real-valued computations, connections to classical FO do at least not seem obvious. Connections to FO over metafinite R-structures, however, seem plausible, and are worth investigating, since there is a close connection between real valued circuits and FO over R-structures.
> 4. Thank you, this is indeed an important paper as it is a precursor for the logical characterisations of Barcelo et al. (ICLR 2020) and Grohe (LICS 2023). We do not think that VVc-GNNs are directly related to our C-GNNs; in our work the aggregation-combine operations are computed by a circuit from some circuit class that delineates the complexity of the aggregation-combine method. In VVc-GNNs the focus is on the format of the message passing, and does not limit complexity of the aggregation-combine method. However, it is an interesting topic of future work to study whether by relating our C-GNNs to variants of VVc-GNNs we may drop the assumption of tail-symmetry. Finally, if the results of Sato et al. are combined with logical characterisations of
> "Hella, Jarvisalo, Kuusisto, Laurinharju, Lempiainen, Luosto, Suomela, Virtema. Weak models of distributed computing, with connections to modal logic. PODC 2012"
> the model of distributed computing that the paper by Sato et al. take inspiration from, one obtains a weaker form of the logical characterisation later proved directly to GNNs by Barcelo. We will add a comment about this to the paper.

---

> ### Comment · Reviewer_FVmx · 2024-08-08
>
> I appreciate the authors' rebuttal, which has addressed many of my concerns, prompting me to increase my score to 6. While the practical impact of the paper remains unclear to me, I am convinced that it has the potential to inspire subsequent research in the field. I would suggest that the authors highlight the takeaways and proofs that could spark future work. Additionally, I would like to remind the authors to **include an official comment on the point raised in item 4 of the rebuttal** as soon as possible.

---

> > ### Author Response · Authors · 2024-08-09
> > **Response by authors**
> >
> > Thank you for your kind comments. We added an official comment that highlights the issues raised in point 4.

---

### Official Review · Reviewer_E5Mq · 2024-07-11

**Soundness:** 4
**Presentation:** 4
**Contribution:** 2
**Rating:** 7
**Confidence:** 5

**Summary:**

In this article, the authors present new contributions on the understanding on the computational framework provided by GNNs.
In particular, they draw a connection between Arithmetic Circuits and GNNs.

Based on a new definition of GNN using arithmetic circuits, they show that the function computed by a GNN on each node can be thought of a function that an (tail-symmetric) arithmetic circuit computes, by stacking the adjacency matrix of the graph and the features vector. In other words, by using arithmetic circuits as comb and aggregation functions inside a message passing framework, the overall procedure can still be thought of as an arithmetic circuit (and vice-versa).

**Strengths:**

- The paper reads well, in particular the introduction about related work is useful. (although the authors may want to include some additional references for completeness, as detailed below in the Questions section).

- The authors propose a new way to think about GNNs, as a correspondence between a given iteration number, and two family circuits (which used to be the COMB and AGG functions).

- I find the several contributions interesting.

**Weaknesses:**

- The main weakness that I can see, is that the definition of the C-GNNs makes the connection between AC-GNN (or close variants, which is what is implemented and used in practice) and circuits. Since the correspondence between GNNs and tail-symmetric arithmetic circuit is established in the following sense:

$$ \text{AC-GNN} \subsetneq (\text{ C-GNNs} = FAC^{0}_{\mathbb{R}^{k}}[\mathcal{A}] )$$

There is potentially a huge gap between AC-GNNs and C-GNNs, that is partially discussed in Remark 3.4. The obscure part of this correspondance seems to arise mainly because of the aggregation functions: i.e., if we have the sum, max and product as aggregation functions, what part of the C-GNNs can we simulate (with polynomial reductions)?

- Also, although I personally find the results interesting, they are not very deep or insightful: if one provide arithmetic circuit power on each node, and arithmetic circuit power as aggregation, then we obtain at most arithmetic circuits.

**Questions:**

- The authors emphasize (and I think it is a good thing) that the computational equivalence they obtain with Arithmetic circuits is something that applies to the message passing framework (the essence of GNNs), regardless of the computational power on the nodes. But in their work, the authors provide an arithmetic circuit as computational power to the nodes. In other words, do the authors believe there is an arithmetic circuit that cannot be computed by an AC-GNN?

- The authors define standard notations like $[n]$, notation for multisets, but not for $(\mathbb{R}^{k})^{*}$. This probably refers to all possible n-tuples of $R^{k}$ for every n. However, it becomes confusing at several locations in the paper.

- From the abstract, the authors mention that their result hold uniformly and non uniformly. This is not mentioned again nor explained in the paper (uniform vs. non uniform) (except if I missed it?)

- The definition of an AC-GNN, as an C-GNN does not coincide with the standard one: a function \sigma^{i} is applied after the combination. This does not align for instance with [2020, Barcelo & Al.].

- Definition 3.3, is unclear: The authors may want to elaborate why the proj_1 and proj_2 operations are made consecutively, why is proj_2 needed for $\sigma^{(i)}$ ? I suppose that this is in order to have access to two different arithmetic consecutively.


- The authors may want to consider in their introduction, in order to compare their result (in particular the independence w.r.t. the activation functions) the references:

- Impact of aggregation function: 		Eran Rosenbluth, Jan Tönshoff, Martin Grohe ``Some might say all you need is sum``
Martin Grohe, Eran Rosenbluth               ``Are Targeted Messages More Effective?``

- Impact of activation function on GNN expressivity: Sammy Khalife ``Graph Neural Networks with polynomial activations cannot express all GC2 queries``

- Sammy Khalife, Amitabh Basu	``On the power of graph neural networks and the role of the activation function.``


- In proof of Theorem 3.11 is used Definition A.7, and mentioned (``via the injective function'', but this definition allows several functions (as long as different depth implies different values taken by the function). Can the authors confirm and/or clarify this?

- I had a look at the proof of Theorem 3.11 in details (I find this result the most interesting). Can the authors confirm: the essence of the proof is to group the input of the initial circuits (vector of reals) and the circuit as a new graph input to the C-GNN? The gates and structure of the initial circuit is then processed by the C-GNN, in order to compensate for the excess (addition and multiplication shown in Algorithm 1). The division performed in Algorithm 1 is allowed as it is the same as multiplying by the inverse of a constant (this number is indeed a constant as it relates to the structure of the circuit, not the input).

**Limitations:**

The authors are transparent in the conclusion of their work about the limitations of their work.

---

> ### Author Rebuttal · Authors · 2024-08-06
>
> ## Regarding the mentioned weaknesses:
> The nature of the relationship between AC-GNNs and C-GNNs lies in the definitions. In Remark 3.4 we assume the aggregation functions to be computable by $\text{FAC}^0_{R^k}$-circuit families, which the max function is not. So to be able to express an AC-GNN that uses max as the aggregation function, a different class of C-GNNs would be needed. The other way around (AC-CNNs simulating C-GNNs) this would boil down to the question which kind of circuit class the circuits used in the C-GNN are part of. For $\text{FAC}^0_{R^k}$ (note: no extra gates) we would intuitively say the AC-GNNs could simulate C-GNNs where every circuit has a restricted depth as the sum or product over n different inputs only takes one gate to compute in a $\text{FAC}^0_{R^k}$ circuit, assuming the circuits are minimal.
>
> While the results may not seem too surprising at first glance, the proofs still involve a lot of technicalities, some mainly attributed to the use of real numbers and functions over those, while others arise from the fact that this essentially shows that message passing does not increase power in this kind of setting.
>
> ## Regarding the questions:
> 1. In their standard definition, arithmetic circuits have access to addition and multiplication gates.
> However, functions which make use of arbitrary multiplication in a meaningful way cannot in general be computed by AC-GNNs of classical definition with constant depth, i.e., with sum as its aggregation and a weighted sum as its combine function.
> If we were to use different aggregation and/or combine functions in the AC-GNN, depending on the chosen functions, we might very well be able to simulate arithmetic circuits using AC-GNNs as well.
> 2. You are correct, we will address this in the revised version.
> 3. From the constructive nature of our proofs, it follows that we will have some notion of uniformity. Defining that notion precisely and proving a formal statement will still require some work.
> 4. Apart from (not) fixing the combination function, our definition aligns with the one mentioned, as $\sigma^{(i)}$ corresponds to the function f that is part of the combination in Barceló et al. We changed the notation here because we refer to this function very frequently as the activation function and therefore wanted it to be written out separately from the combination function.
> 5. In our current definition, $\mathcal{N}(i)$ is a function pointing to a CGNN basis for every layer, which in turn is a tuple $(\mathcal{S}, \mathcal{A})$ containing a set of functions computed by a circuit family $\mathcal{S}$ and a set of activation functions $\mathcal{A}$.
> We needed the projections to refer to the individual functions $\mathcal{C}^{(i)}$ and $\sigma^{(i)}$ for layer $i$. We will improve the somewhat poor presentation of this.
> 6. Thank you, we will look into these.
> 7. This is indeed a misleading formulation. Any numbering satisfying the condition of Definition A.7 would suffice, we will clarify this.
> 8. Yes, this is indeed the essence of the proof. The details certainly require a bit of technical work, but this portrays the general idea well.

---

> > ### Comment · Reviewer_E5Mq · 2024-08-09
> >
> > Thanks for the detailed explanations. I think this paper will be of interest for the GNN community and researchers at the interface with circuit complexity. I strongly encourage the authors to include the changes discussed and will maintain my score as is.

---

### Official Review · Reviewer_3Xib · 2024-07-12

**Soundness:** 2
**Presentation:** 3
**Contribution:** 3
**Rating:** 6
**Confidence:** 2

**Summary:**

This paper provides a new lens through which the expressive power of GNNs is analyzed via arithmetic circuits and derives expressiveness limits for general GNNs.

**Strengths:**

- this paper is indeed modular, well-written, and cleanly organized
- it introduces an interesting perspective for analyzing GNN expressiveness

**Weaknesses:**

- I think this paper would benefit from more detailed discussion on the relationship with WL tests.

**Questions:**

- Is there any numerical experiments you can run to demonstrate your theoretical findings?

**Limitations:**

The limitations have been sufficiently addressed in the paper.

---

> ### Author Rebuttal · Authors · 2024-08-06
>
> **I think this paper would benefit from more detailed discussion on the relationship with WL tests.**
> The Weisfeiler Lehman tests (and its variants) usually refer to the problem of graph isomorphism, and give insight to the question whether two graphs can be distinguished by some GNN model. Since we are concerned with comparing the classes of functions over the reals that can be computed (using distributed computation/message passing in the case of C-GNNs and AC-GNNs), the WL test does not precisely relate to our main focus. We of course mention WL tests as a well understood technique to compare GNN expressivity.
>
> **Is there any numerical experiments you can run to demonstrate your theoretical findings?**
> As it currently stands, there does not seem to be any meaningful experiments that could be run for this model of computation. Essentially, the problem is that not much is known about the computational power of the circuit classes we utilize. Further research into functions computable by practically implementable arithmetic circuits of different sizes and depths would be required.
> This could e.g. be done by investigating the Boolean parts of different complexity classes defined by families of arithmetic circuits, i.e., the classes when restricted to Boolean inputs. This research would pinpoint properties that would be provably outside of the capabilities of particular GNN models, whose learnability could be then tested in practice. We believe that our results will motivate research for these circuit classes.

---

> ### Comment · Reviewer_3Xib · 2024-08-08
>
> Many thanks for your rebuttal!

---

### Author Response · Authors · 2024-08-09
**Connections between our work and VVc-GNNs**

As requested by the reviewer FVmx, we highlight here possible connections and differences of our work and VVc-GNNs introduced in [1] that we mentioned in our response to FVmx.

The article [1] is indeed an important paper as it is a precursor for the logical characterisations of Barceló et al. [2] and Grohe [3]. We do not think that VVc-GNNs are directly related to our C-GNNs; in our work the aggregation-combine operations are computed by a circuit from some circuit class that delineates the complexity of the aggregation-combine method. In VVc-GNNs the focus is on the format of the message passing, and does not limit complexity of the aggregation-combine method. However, it is an interesting topic of future work to study whether by relating our C-GNNs to variants of VVc-GNNs we may drop the assumption of tail-symmetry. Finally, if the results of Sato et al. are combined with logical characterisations of [4] (where the model VVc of distributed computing that [1] take inspiration from is studied) one obtains a weaker form of the logical characterisation later proved directly for GNNs in [2]. We will add a comment about this in the final version.

[1] Ryoma Sato, Makoto Yamada, Hisashi Kashima: Approximation Ratios of Graph Neural Networks for Combinatorial Problems. NeurIPS 2019.

[2] Pablo Barceló, Egor V. Kostylev, Mikael Monet, Jorge Pérez, Juan Reutter, Juan Pablo Silva. The Logical Expressiveness of Graph Neural Networks. ICLR 2020.

[3] Martin Grohe. The Descriptive Complexity of Graph Neural Networks. LICS 2023

[4] Lauri Hella, Matti Järvisalo, Antti Kuusisto, Juhana Laurinharju, Tuomo Lempiäinen, Kerkko Luosto, Jukka Suomela, Jonni Virtema. Weak Models of Distributed Computing, with Connections to Modal Logic. PODC 2012.

---

### Decision · Program_Chairs · 2024-09-25

**Decision:**

Accept (poster)

**Comment:**

All reviewers agreed that this paper provides an important theoretical contribution by connecting arithmetic circuits and GNNs in a novel way. All major concerns could be resolved during the rebuttal. Therefore, I recommend acceptance of the paper as a poster.